# Robust Optimization and Power Management of a Triple Junction Photovoltaic Electric Vehicle with Battery Storage

**DOI:** 10.3390/s22166123

**Published:** 2022-08-16

**Authors:** Salah Beni Hamed, Mouna Ben Hamed, Lassaad Sbita, Mohit Bajaj, Vojtech Blazek, Lukas Prokop, Stanislav Misak, Sherif S. M. Ghoneim

**Affiliations:** 1Physic Department, High School of Engineers of Tunis, Tunis 1008, Tunisia; 2Electrical Department, National Engineering School of Gabes, Gabes 6029, Tunisia; 3Department of Electrical Engineering, Graphic Era (Deemed to be University), Dehradun 248002, India; 4ENET Centre, VSB—Technical University of Ostrava, 708 00 Ostrava, Czech Republic; 5Electrical Engineering Department, College of Engineering, Taif University, P.O. Box 11099, Taif 21944, Saudi Arabia

**Keywords:** PV generator, triple junction, first order sliding mode, MPPT, nonlinear control, electric vehicle, DC-DC power converters, energy management

## Abstract

This paper highlights a robust optimization and power management algorithm that supervises the energy transfer flow to meet the photovoltaic (PV) electric vehicle demand, even when the traction system is in motion. The power stage of the studied system consists of a triple-junction PV generator as the main energy source, a lithium-ion battery as an auxiliary energy source, and an electric vehicle. The input–output signal adaptation is made by using a stage of energy conversion. A bidirectional DC-DC buck-boost connects the battery to the DC-link. Two unidirectional boost converters interface between the PV generator and the DC link. One is controlled with a maximum power point tracking (MPPT) algorithm to reach the maximum power points. The other is used to control the voltage across the DC-link. The converters are connected to the electric vehicle via a three-phase inverter via the same DC-link. By considering the nonlinear behavior of these elements, dynamic models are developed. A robust nonlinear MPPT algorithm has been developed owing to the nonlinear dynamics of the PV generator, metrological condition variations, and load changes. The high performance of the MPPT algorithm is effectively highlighted over a comparative study with two classical P & O and the fuzzy logic MPPT algorithms. A nonlinear control based on the Lyapunov function has been developed to simultaneously regulate the DC-link voltage and control battery charging and discharging operations. An energy management rule-based strategy is presented to effectively supervise the power flow. The conceived system, energy management, and control algorithms are implemented and verified in the Matlab/Simulink environment. Obtained results are presented and discussed under different operating conditions.

## 1. Introduction

With the fast growth of cars, especially car ownership, the number of vehicles in the world increases day by day [1,2]. This has led to an important rise in oil consumption in the transport sector [3,4,5]. As more of the vehicle’s energy is obtained by an internal combustion engine, the carbon dioxide (CO_2_) emissions will increase [6,7,8]. Nowadays, the CO_2_ rate has crossed 400 ppm and will increase. Faced with the energy crisis, climate change, and the need to save the earth and people’s lives, the development of a new vehicle structure is considered by looking for some sustainable technologies that reduce energy consumption or utilize renewable and clean energy sources [9,10]. Other energy sources are the challenge of most proposed solutions.

To sufficiently reduce both consumption and transportation emissions, electric vehicles are considered an effective transport tool. Two kinds of electric vehicles have been discovered: pure electric vehicles and hybrid vehicles [11,12,13].

Electric vehicles cover more than a research field. The structure, benefits, and drawbacks of each are thoroughly studied in b. A large number of studies focused on the vehicle’s traction [14,15,16].

The main objectives in the motor choice for the traction part are to increase the electric vehicle’s performance while minimizing both the vehicle weight and energy consumption. Many research studies are carried out with this aim [17,18,19]. In Reference [17], many kinds of electric machines are used for electric vehicles, and the importance of choosing the traction machines is highlighted in this case. Besides, the number of motors and their placement in the vehicle are also discussed [20].

Power management in electrical vehicles was also discussed and research reviews and studies were conducted [21,22,23]. Among these approaches, we find dynamic programming strategy [21], all or nothing strategy [24], rule-based strategy, including deterministic and fuzzy logic rules [25,26], filtration strategy [27], and predictive model strategy [28].

Another interesting research field in electric vehicles concerns vehicle autonomy [29,30]. Suggested solutions for improving electric vehicle autonomy can be divided into two kinds. Some of them concentrate on working on battery technology. The other option is to use rechargeable batteries [31]. Using PV energy sources is one of the suggested recharge battery solutions [32]. The PV panels are located on the body of the vehicle. The recharge time is improved by using the PV generator at its maximum power. For this goal, different MPPT algorithms are suggested in the literature: perturb and observe (P & O) [33,34,35], the incremental inductance (IC) [36,37,38], fuzzy logic (FL) [39,40,41], neuronal networks (NN) [42,43,44], particle swarm optimization (PSO) [45,46,47], and sliding mode [48,49] etc. High oscillation remains the major weakness of these MPPT approaches.

According to a literature review, the used PV generator is designed with mono-junction solar cells. The major drawback of this technology is its low efficiency. This fact led to increasing the PV panel covered area.

The aim of this paper is to improve electric vehicle performance. To minimize the overall vehicle weight, a highly efficient PV generator based on multi-junction solar cell technology is conceived. A lithium-ion battery bank storage system is used. A nonlinear robust sliding mode-based MPPT algorithm and a Lyapunov function-based nonlinear control approach for DC-DC converters with an energy management rules-based approach are being investigated as solutions to improve the battery recharge time.

This paper is structured as follows. In Section 2, the forward simulation model of the electric vehicle powertrain system is established. Section 3 presents the energy management and control approaches. In Section 4, the simulation results are presented and discussed. Conclusions and some suggested prospects are provided in the Section 5.

## 2. Modeling of the PV Electric Vehicle Powertrain System

The structure of the used PV electric vehicle powertrain system is shown in Figure 1.

### 2.1. Modeling of Triple-Junction Solar Cell InGap/InGaAs/Ge

The triple-junction InGap/InGaAs/Ge solar cell includes three sub-cells with different wavelengths in series.

Electrical representation, by adapting the decreased energy band-gap from the top to the bottom structure, is given in Figure 2.

Based on Figure 1, the solar cell current can be written as follows in Equation (1).
(1)$$I_{pv}=I_{pi}-I_{Di}-I_{Rshi}$$

The index *i* equals 1 for the top sub-cell. For the medium sub-cell, *I* = 2, and for the bottom sub-cell, *I* = 3.

The light generated current is given by
(2)$$I_{pi}=\frac{G}{G_{STC}}\left[I_{sccSTCi}+\gamma \left(T-T_{STC}\right)\right]$$
where *T_STC_* is the temperature solar cell at standard test conditions in °C, *T* is the temperature solar cell in °C, *G* and *G_STC_* are the solar radiation and the solar radiation at standard test conditions in w/m^2^, respectively, $I_{sccSTCi}$ is the short circuit current at standard test conditions, and γ is the temperature coefficient of the actual short circuit current in A/°C.

The diode current intensity is expressed as in Equation (3).
(3)$$I_{Di}=I_{0i}\left[\exp(\frac{qU_{Di}}{n_{i}BT})-1\right]$$

Its voltage equation is given in Equation (4).
(4)$$U_{Di}=U_{pvi}+R_{si}I_{pv}$$

The diode saturation current $I_{0i}$ is expressed as Equation (5)
(5)$$I_{0i}=K_{i}\;T(\frac{3+\delta_{i}}{2})\exp(\frac{-E_{BGi}}{n_{i}BT})$$
where q is the electric charge of an electron, $n_{i}$ is the ideality factor of a diode, $E_{BGi}$ is the band-gap energy, B is the Boltzmann’s constant, and $K_{i}$ and $\delta_{i}$ are constant.

The energy band-gap is given in Equation (6).
(6)$$E_{BGi}(T)=E_{BGi}(0)+(\frac{\alpha_{i}T²}{T+\beta_{i}})$$

With $\alpha_{i}$ is a material energy per Kelvin fitting parameters and $\beta_{i}$ is a material temperature fitting parameters.

By using Equations (3) and (4), the triple-junction solar cell $I_{pv}(U_{pv})$ characteristic is obtained.
(7)$$\begin{array}{l} U_{pv}=\frac{n_{1}BT}{q}Ln(\frac{I_{p1}-I_{pv}-I_{Rsh1}}{I_{sat1}}+1)+\frac{n_{2}BT}{q}Ln(\frac{I_{p2}-I_{pv}-I_{Rsh2}}{I_{sat2}}+1)\\ +\frac{n_{3}BT}{q}Ln(\frac{I_{p3}-I_{pv}-I_{Rsh3}}{I_{sat3}}+1)-R_{s}I_{pv} \end{array}$$

$R_{s}$ is the equivalent of serial resistance. It is denoted by Equation (8)
(8)$$R_{s}=\sum_{j=1}^{3}R_{si}$$

Based on the load demand, a suitable triple junction PV generator is conceived. According to the Matlab/Simulink test of the established model, the electric characteristic curves are obtained, as shown in Figure 3. From these characteristics, we note that for each pair of radiation and temperature, there is one operating point in which the generated power is at its maximum value. Moreover, in general, the meteorological are intermittent. As a result, the power produced may differ from the power demanded. Therefore, an MPPT algorithm seems to be the most suitable solution to extract the maximum power on the one hand. On the other hand, to better manage the energy flow and protect the system, an energy management strategy has to be integrated.

### 2.2. Modeling of Electric Vehicle Dynamics

Due to the multiple performance of the three-phase permanent magnet synchronous motor (PMSM), it is used in the monitoring part of the electric vehicle. The dynamic electrical behavior of the three-phase PMSM can be represented as space vectors by the following nonlinear equations established in the (d, q) frame as [50].
(9)$$\left\{ \begin{array}{l} U_{sd}=R_{s\;}i_{sd}+L_{sd}\frac{di_{sd}}{dt}-\omega_{s}L_{sq}i_{sq}\\ U_{sq}=R_{s}\;i_{sq}+L_{sq}\frac{di_{sq}}{dt}+\omega_{s}L_{sd}i_{sd}+\omega_{s}\phi_{a} \end{array}\right.$$
where $U_{sd}$, $U_{sq}$ and $i_{sd}$*,*
$i_{sq}$ direct and quadratic stator voltage and current components, respectively. $R_{s}$ is the stator resistance. $L_{sd}$ and $L_{sq}$ are the direct and reverse self-stator inductance components, $\omega_{s}$ is the rotor angular speed and $\phi_{a}$ represents the permanent magnet flux linkage.

Its mechanical behavior is as follows.
(10)$$J_{t}\frac{d\omega_{s}}{dt}=n_{p}\left(T_{em}-T_{r}\right)-f_{vf}\omega_{s}$$

With $n_{p}$ is the pair poles’ number, $f_{vf}$ is the coefficient of the viscous friction, $J_{t}$ is the total moment of inertia $T_{em}$ is the electromagnetic torque, and $T_{r}$ is the resistant torque.

By neglecting the influence of the vehicle’s lateral and vertical dynamics, $T_{r}$ is expressed as follows.
(11)$$T_{r}=F_{r}\;R_{t}$$
where $F_{r}\;$ is the total resistive force and $R_{t}$ is the radius of the vehicle’s tire.

The force is given by
(12)$$F_{r}=F_{rr}+F_{ar}+F_{sr}$$

In which $F_{rr}$ is the rolling resistance, $F_{ar}$ is the air resistance, and $F_{sr}$ is the slope resistance.

The forces $F_{rr}$, $F_{ar}$ and $F_{sr}$ are expressed as Equations (13)–(15), respectively.
(13)$$F_{rr}=M_{v}\;g\;f_{rr}$$
(14)$$F_{ar}=\frac{1}{2}\rho_{a}A_{fa}C_{ad}V_{v}²$$
(15)$$F_{sr}=M_{v}\;g\sin(\alpha_{rs})$$

In which $M_{v}$ is the total mass of the vehicle, g is the gravity acceleration, $f_{rr}$ is the coefficient of the rolling resistance, $\rho_{a}$ is air density, $A_{f}{}_{a}$ is the frontal surface area of the vehicle, $C_{ad}$ is the aerodynamic drag coefficient, $V_{v}$ is the speed of the vehicle, and $\alpha_{rs}$ is the street inclination angle.

### 2.3. Modeling of the Lithium-Ion Battery

Since the photovoltaic system’s electrical characteristics depend on intermittent weather conditions, its output energy may be insufficient to meet the load demands. Storing energy seems to be necessary. The battery is the most commonly used storage system in a standalone system [29]. Lithium-ion batteries are chosen as a suitable storage system for electric vehicles due to their power density, high specific energy, and long life expectancy.

In the existing literature, various lithium-ion battery models have been developed [51]. The most commonly used is the one developed with Shepherd [52]. The extended modified Shepherd model is represented with a controlled voltage source and an internal resistance, as indicated by Equation (16).
(16)$$U_{sb}=E_{sb}-Ri_{n}\;I_{sb}$$

With $U_{sb}$ is the battery voltage, $E_{sb}$ is the controlled voltage source, $Ri_{n}\;$ is the internal battery resistance, and $I_{sb}$ is the battery current.

For charging mode, we have:(17)$$E_{sb}(Q_{a},i_{sbf})=E_{sb0}-K_{sb}\frac{Q_{n}}{Q_{n}-Q_{a}}\;i_{sbf}-K_{sb}\frac{Q_{n}}{Q_{n}-Q_{a}}\;Q_{a}+A_{b}\;\exp(-B_{b}t)$$

In discharge mode, we can write:(18)$$E_{sb}(Q_{a},i_{sbf})=E_{sb0}-K_{sb}\frac{Q_{n}}{Q_{n}+0.1Q_{a}}\;i_{sbf}-K_{sb}\frac{Q_{n}}{Q_{n}-Q_{a}}\;Q_{a}+A_{b}\;\exp(-B_{b}t)$$
where $E_{sb}$ is the no-load voltage, $E_{sb0}$ is the battery constant voltage, $Q_{n}$ and $Q_{a}$ are nominal and available battery capacities, $i_{sbf}$ is the low frequency component of the battery current, $K_{b}$ is the polarization voltage, $A_{b}$ is the battery exponential zone amplitude, and $B_{b}$ is the battery exponential zone time constant inverse.

The available battery capacity is defined as
(19)$$Q_{a}=\int i_{sb}dt$$

Here, $i_{sb}$ is the battery current.

At any given time, the available charge of the battery is expressed over the battery state of charge (SOC). It is defined as
(20)$$SOC(t)=SOC_{in}-\frac{1}{Q_{n}}\int_{0}^{\tau }i_{sb}(\tau )d\tau$$

The initial voltage of the battery depends on the state of charge [53].
(21)$$E_{sb0}(SOC_{in})=a_{0}+a_{1}Ln(SOC_{in})+a_{2}Ln(1-SOC_{in})+\frac{a_{3}}{SOC_{in}}+a_{4}SOC_{in}$$
where $a_{0}\ldots a_{4}$ are parameters to fit the model to a specific battery and $SOC_{in}$ is the initial battery state of charge.

The battery power is computed as follows.
(22)$$P_{sb}=U_{sb}\;i_{sb}$$

### 2.4. Modeling of the DC-DC Converters

#### 2.4.1. Modeling of the Unidirectional DC-DC Boost Converter

Two cascading DC-DC boost converters are integrated between the main triple-junction PV generator and the electric vehicle. The first is used to track the maximum power points. The other is utilized to adapt the low DC voltage to the desired DC-link inverter voltage. The unidirectional DC-DC boost converter is a suitable configuration in this phase. According to Figure 4, the boost converter is composed of a high frequency coil (L_f_), an IGBT transistor (T_1_), a diode D_ar_, and C_f_ as an output voltage filter.

In this chopper, there is an on–off switch (T_1_). The working principle depends on the state of this switch.

When the switch T_1_ is on, the source current follows the inductor and the switch. Only the capacity supplies the load. At this stage, the inductor stores energy, and the capacity discharges energy through the load.

When the switch T_1_ is off, the diode D_ar_ will be ready to conduct. At this phase, the inductor loses the stored energy for charging the capacitor.

The switch is controlled by using a pulse width modulation signal $S_{a}$.

A bilinear average switching model is obtained by considering some idealities and taking into account the nature of the switch,
(23)$$\left\{ \begin{array}{l} \frac{di_{in}}{dt}=\frac{U_{in}}{L_{f}}-(1-\mu_{c1})\frac{U_{dc}}{L_{f}}\\ \frac{dU_{dc}}{dt}=(1-\mu_{c1})\frac{i_{in}}{C_{f}}-\frac{i_{dc}}{C_{f}} \end{array}\right.$$
where $U_{in}$, $U_{dc}$, and $i_{in}$, $i_{dc}$ are, respectively, the input voltage, the DC-link voltage, the input, and the output currents of the boost converter, and $u_{c1}$ is the averaged value of the pulse width modulation signal u.

#### 2.4.2. Modeling of the Bidirectional DC-DC Buck-Boost Converter

The battery is the main storage system in this application. It behaves as a bidirectional system. To manage the energy transfer, the battery is connected to the DC-link by means of two quadrant DC-DC converters. The most commonly used in this stage is the bidirectional DC-DC buck-boost converter.

Referring to Figure 5, the buck-boost is composed of a high frequency coil (*L*_2_) and two IGBT switches, *T*_2_ and *T*_3_.

When the switch T_2_ and the diode D_3_ states are on, the battery provides energy to the load. In this case, the bidirectional chopper works in the boost operating mode.

Now, let us consider the case when the switch T_3_ and the diode D_2_ are in conduction. In this case, the battery current is negative and the battery charges.

To distinguish the operating mode, a binary variable m is defined. Thus, we can write:(24)$$m=\left\{ \begin{array}{l} \begin{array}{cc} 1 & i_{sbref}>0\;\mathrm{boost}\;\mathrm{mode} \end{array}\\ \begin{array}{cc} 0 & i_{sbref}<0\;\mathrm{buck}\;\mathrm{mode} \end{array} \end{array}\right.$$
where $i_{sbref}$ is the target battery current.

Hence, the bidirectional buck-boost converter average model is given by the following Equation (25).
(25)$$\left\{ \begin{array}{l} \frac{di_{sb}}{dt}=-[m(1-\mu_{c2})+(1-m)\mu_{c3}]\frac{U_{dc}}{L_{2}}+\frac{U_{sb}}{L_{2}}\\ i_{dc2}=[m(1-\mu_{c2})+(1-m)\mu_{c3}]i_{sb} \end{array}\right.$$

By combining the two operating modes, a virtual control signal is defined. It is designed with $m_{23}$. The latter is expressed by
(26)$$m_{23}=m(1-\mu_{c2})+(1-m)\mu_{c3}$$

The converter model becomes
(27)$$\left\{ \begin{array}{l} \frac{di_{sb}}{dt}=-m_{23}\frac{U_{dc}}{L_{2}}+\frac{U_{sb}}{L_{2}}\\ i_{dc2}=m_{23}\;i_{sb} \end{array}\right.$$

The DC-link feeding the three-phase inverter is modeled by the DC voltage at the output of the filter capacitor. It is represented by the following Equation (28).
(28)$$C_{f}\frac{dU_{dc}}{dt}=(1-u_{c1})i_{in}+m_{23}i_{sb}-i_{dc}$$

## 3. Control Approaches and Energy Management Strategy

### 3.1. MPPT Algorithms

#### 3.1.1. P & O Algorithm

Thanks to its ease of implementation and its simplicity, the P & O algorithm is the most commonly used [54]. As its name suggests, it is based on the disturbing the PV system and then observing the future impact of the added disturbance on the PV generator. In fact, if the reference voltage is disturbed in such a direction, the power of the PV generator increases. This means that disturbing the PV system moves its operating point to the maximum power point (MPP). Therefore, in this case, the P & O algorithm kept going, disturbing the reference voltage in the same direction. However, when the system power decreases, this means that disturbing the reference voltage moves the operating point far away from its optimal one. Then, the P & O reverses the sign of the added perturbation. This working principle is repeated until the MPP is reached. Since this algorithm perturbs the operating point of the PV system, its terminal power will fluctuate around the MPP, although solar radiation and temperature are constant leading to a power loss in the system. The flowchart of the P & O algorithm is given in Figure 6.

#### 3.1.2. Fuzzy Logic Algorithm

Fuzzy logic (FL) is a numerical computational approach. This concept was first introduced by Lotfi Zadeh in 1965 [55]. It is based on the fuzzy set theory. As no mathematical model is needed for this approach and a human decision-making concept is used, this strategy may give highly effective results. FL strategy can be a challenge for PV systems as reported in Reference [39]. In the presented work, a mamdani type fuzzy system is used for the FL MPPT approach. The error defined with the PV generator power variation over the PV generator voltage and the change of the error over time are chosen as the FL system inputs. The FL algorithm provides, at its output, the change of the DC-DC boost converter duty cycle. The mathematical expressions of the FL system input and output variables are given with Equations (29) and (30), respectively.
(29)$$e(k)=\frac{P_{pv}(k)-P_{pv}(k-1)}{V_{pv}(k)-V_{pv}(k-1)}$$
(30)$$\Delta e(k)=\frac{\varepsilon (k)-\varepsilon (k-1)}{T_{s}}$$
where $T_{s}$ is the sample time.

To implement the mamdani FL system, four steps are to be followed, as illustrated in Figure 7.

At the fuzzification step, the real input and output signals are converted to fuzzy sets. For each input and output variables, there were seven membership functions, as represented in Figure 8a–c, respectively. Triangular and trapezoidal types of membership functions are chosen in this work. This choice is based on the trial and error method. In fact, many repetitive tests are done until suitable results are obtained. The linguistic variables BN, MN, SN, Z, SP, MP, and BP indicate big negative, medium negative, small negative, zero, small positive, medium positive, and big positive, respectively. The obtained fuzzy input and output variables are then treated by an inference engine. In this work, a sum-prod inference algorithm is used. The inputs are mapped to the outputs by using the if-then rules as indicated in Table 1. The number of the rules is fixed based on the membership function number. The fuzzy output variables obtained at the inference engine step are then converted into a crisp value. In this application, a centroid defuzzification approach is used. The real duty cycle to be applied to the real system is defined by a recurrent equation, as illustrated with Equation (31).
(31)u(k)=u(k−1)+NΔu(k)
where *N* is an adjustable positive gain. 

#### 3.1.3. Sliding Mode MPPT-Based Algorithm

The main objective of the MPPT algorithm is to extract the maximum power, on the one hand. On the other hand, despite changes in meteorological conditions and system parameters, the system operating points must remain optimal. Since the photovoltaic system is highly nonlinear, a sliding mode (SM)-based MPPT approach is conceived.

Achieving the maximum power extraction using the sliding mode strategy entails the suitable choice of the sliding mode surface. For PV systems, the alteration of the PV generator power over its current is equal to zero. To this end, the dynamic of the PV generator’s power needs to be identified. Deriving the power over the current, we get
(32)$$\frac{\partial P_{pv}}{\partial I_{pv}}=I_{pv}\left(\frac{\partial U_{pv}}{\partial I_{pv}}+\frac{U_{pv}}{I_{pv}}\right)$$

To carry out the maximum power extraction using the sliding mode approach, a suitable choice of the sliding mode switching function is as follows.
(33)$$H(t,U_{pv},I_{pv})=\frac{\partial U_{pv}}{\partial I_{pv}}+\frac{U_{pv}}{I_{pv}}$$

Since at the maximum power point, the sliding mode function $H(t,U_{pv},I_{pv})$ converges to zero, the maximum power extraction in a PV generator can be formulated as an optimization problem which minimizes the sliding mode switching function while satisfying the inequality constraints.

The objective function and the associated constraints of the optimization problem can be formulated as follows:(34)$$J=\min\left\{H(t,U_{pv},I_{pv})\right\}$$

During the operating conditions of the vehicle, the following constraints are to be satisfied:(35)$$0\leq P_{m}\leq P_{nom}$$
(36)$$0\leq U_{m}\leq U_{nom}$$
(37)$$0\leq I_{m}\leq I_{nom}$$
where $P_{m}$ and $P_{nom}$ are the motor power and its nominal value, respectively. $U_{m}$, $U_{nom}$ are the motor voltage and its nominal value, respectively. $I_{m}$ and $I_{nom}$ design the motor’s current and its nominal value, respectively.

A dynamic tracker based on a DC-DC converter is used to track the maximum power points.

Let us design with u(t) the control law of the DC-DC boost converter. By using the first order sliding mode control based on the equivalent control approach [56,57], u(t) is defined as follows.
(38)$$u(t)=u_{eq}(t)+u_{d}(t)$$
where $u_{eq}(t)$ is the equivalent duty cycle and $u_{d}(t)$ is the discontinuous term.

Assuming that the ideal sliding mode is established, we get
(39)$$u_{eq}(t)=1-\frac{U_{pv}}{U_{in}}$$

Its discontinuous term is given as
(40)$$u_{d}(t)=-U\;sign(H(t,U_{pv},I_{pv}))$$

The control signal is a duty cycle, so the real control law signal of the boost converter is defined by the following set of equations.
(41)$$u(t)=\{\begin{array}{ccc} 0 & & if\;u(t)\leq 0\;\\ 1-\frac{U_{pv}}{U_{in}}-U\;sign(H(t,U_{pv},I_{pv})) & & if\;0<u(t)<1\\ 1 & & if\;u(t)\geq 1 \end{array}$$

#### 3.1.4. Comparative Study

In order to highlight the effectiveness and the robustness of the conceived sliding mode MPPT algorithm, PV system consisting of a triple-junction PV generator, a DC-DC buck converter and a resistive load were implemented in Matlab/Simulink platform. Different operating modes were simulated in which the investigated MPPT algorithm was evaluated in comparison with the P & O and fuzzy logic MPPT algorithms under the same test conditions. Three operating conditions, including simultaneous abrupt variation both in radiation and temperature, abrupt load variation and the case of simultaneous abrupt variation in radiation, temperature, and load were considered. Obtained results are shown in Figure 9. The used radiation, temperature, and load trajectories are shown in Figure 9a–c, respectively. Figure 9d presents the duty cycles with the three MPPT algorithms. The evolution of the PV generator power for the three MPPT algorithms is depicted in Figure 9d. From *t* = 1s to *t* = 2 s, abrupt radiation and temperature variations are highlighted. In fact, for this case, the load is fixed to 5 Ω. The radiation and temperature levels are fixed at first to 900 w/m^2^ and 27 °C, respectively. At *t* = 1 s, both radiation and temperature increase, respectively, to 100 w/m^2^ and 77 °C. At *t* = 2 s, both radiation and temperature levels are maintained as constant at their previous values. The load increase to 10 Ω at *t* = 2 s and to 15 Ω at *t* = 3 s. In the case of simultaneous abrupt radiation, temperature and load variation, all these variables are simultaneously changed. In fact, at *t* = 4 s, the radiation, temperature, and load increase to 1100 w/m^2^, 87 °C, and 8 Ω, respectively. Since *t* = 5 s, radiation is fixed to 1050 w/m^2^, the temperature decreases to 52 °C and the load increases to 12 Ω.

To judge the performance of the conceived algorithm, different parameters, including response time, tracking error, objective function value, stabilization time, and voltage loss, are used. Based on these criteria, a comparison of these MPPT algorithms is extracted and grouped in Table 2. The sliding mode MPPT algorithm has a quick dynamic performance for the three operating modes in comparison with the P & O and fuzzy logic MMPT algorithms. The smaller value time for both response time and stabilization time factors are obtained with the sliding mode MPPT algorithm, as depicted in Table 2. Besides, the smaller value for tracking error and objective function value is assumed with the sliding mode MPPT algorithm. Thus, we conclude that the sliding mode MPPT algorithm remains the most precise compared to other algorithms. Moreover, small loss voltage value was obtained for both abrupt radiation, temperature, and load variation cases, proving the high performance of the investigated sliding mode MPPT algorithm as shown in the obtained simulation results and over the computed values stored in Table 2. For these reasons, the conceived first order sliding mode MPPT algorithm seems to be the best choice to be used in photovoltaic electric vehicle.

### 3.2. Field Oriented Control Strategy

A field-oriented control (FOC) approach is used to independently control the PMSM flux and the torque and improve the dynamic performance. The most commonly used scheme of the FOC strategy includes two inner loops for the direct and reverse stator current control and one outer loop for the speed regulation. The target value of the reverse current component is delivered at the speed output controller. The desired value of the direct stator current component is either fixed at zero value or to a computed value from the high-speed control strategy depending on the operating mode. Working at a high-speed region is assumed thanks to the field-weakening region algorithms [50]. A strategy based on the maximum torque per ampere (MTPA) is used. Figure 6 denotes the principle of the field-weakening algorithm (Figure 10).

### 3.3. Nonlinear Control

Combining Equations (23), (27) and (28), the following bilinear switched model of the global system is expressed as Equation (42).
(42)$$\left\{ \begin{array}{l} \frac{di_{in}}{dt}=\frac{u_{in}}{L_{f}}-(1-u_{c1})\frac{u_{dc}}{L_{f}}\\ \frac{di_{sb}}{dt}=-m_{23}\frac{u_{dc}}{L_{2}}+\frac{u_{sb}}{L_{2}}\\ \frac{du_{dc}}{dt}=(1-u_{c1})\frac{i_{in}}{C_{f}}+m_{23}\frac{i_{sb}}{C_{f}}-\frac{i_{dc}}{C_{f}} \end{array}\right.$$

At the average model Equation (45) over the switching periods, we get
(43)$$\left\{ \begin{array}{l} \frac{dI_{in}}{dt}=\frac{U_{in}}{L_{f}}-(1-\mu_{c1})\frac{U_{dc}}{L_{f}}\\ \frac{dI_{sb}}{dt}=-M_{23}\frac{U_{dc}}{L_{2}}+\frac{U_{sb}}{L_{2}}\\ \frac{dU_{dc}}{dt}=(1-\mu_{c1})\frac{I_{in}}{C_{f}}+M_{23}\frac{I_{sb}}{C_{f}}-\frac{I_{dc}}{C_{f}} \end{array}\right.$$
where $I_{in}$ is the average value of $i_{in}$, $U_{dc}$ is the average value of DC-link voltage $u_{dc}$, $U_{sb}$ is the average value of the battery voltage $u_{sb}$, $I_{dc}$ is the average value of the load current, and $\mu_{c1}$ and $M_{23}$ are the DC-DC converter duty cycles.

The obtained model is a multi-input, multi-output system. Moreover, it is highly nonlinear. Therefore, a nonlinear control-based Lyapunov approach, as mentioned in Reference [55], is used. One of the control objectives is to enforce the DC-link voltage u_dc_ to track its target reference value *U_dcref_*, despite external and internal disturbances. An indirect control strategy is used to cope with this problem. It is based on the control current. Based on the power input equals to power output (PIPO) principle, the desired input current of the DC-DC boost converter at the DC-link *I_inref_* is expressed as
(44)$$I_{inref}=f_{i}(\frac{U_{dcref\;}I_{dc}-U_{sb}I_{sb}}{U_{in}})$$
where $f_{i}\geq 1$ is an ideality factor representing all losses.

Let us design with $\varepsilon_{1}$ and $\varepsilon_{2}$ the DC-DC boost converter input current error, and the DC-link voltage error, respectively. They are expressed as Equation (45).
(45)$$\left\{ \begin{array}{l} \varepsilon_{1}=I_{in}-I_{inref}\\ \varepsilon_{2}=U_{dc}-U_{dcref} \end{array}\right.$$

Deriving Equation (45), we get
(46)$$\left\{ \begin{array}{l} {\dot{\varepsilon }}_{1}=\frac{U_{in}}{L_{f}}-(1-\mu_{c1})\frac{U_{dc}}{L_{f}}-{\dot{I}}_{inref}\\ {\dot{\varepsilon }}_{2}=(1-\mu_{c1})\frac{I_{in}}{C_{f}}+M_{23}\frac{I_{sb}}{C_{f}}-\frac{I_{dc}}{C_{f}}-{\dot{U}}_{dcref} \end{array}\right.$$

To enforce that the DC-link voltage regulation is assumed with the current and vice versa, the derivative time of $\varepsilon_{1}$ and $\varepsilon_{2}$ are forced to a specific equation.
(47)$$\left\{ \begin{array}{l} {\dot{\varepsilon }}_{1}=-d_{1}\varepsilon_{1}+\varepsilon_{2}\\ {\dot{\varepsilon }}_{2}=-d_{1}\varepsilon_{2}-\varepsilon_{1} \end{array}\right.$$

Using Equations (46) and (47), the control law of the DC-link boost converter is obtained in Equation (48).
(48)$$\mu_{c1}=1-\frac{L_{f}}{U_{dc}}\left[d_{1}\varepsilon_{1}-\varepsilon_{2}+\frac{U_{in}}{L_{f}}-{\dot{I}}_{inref}\right]$$

Let us design with $\varepsilon_{3}$ the regulation error of the battery current.
(49)$$\varepsilon_{3}=I_{sb}-I_{sbref}$$
where $I_{sbref}$ is the desired value of the battery current generated from the proposed energy management algorithm.

Its time derivative is defined as Equation (50).
(50)$${\dot{\varepsilon }}_{3}=-M_{23}\frac{U_{dc}}{L_{2}}+\frac{U_{sb}}{L_{2}}-{\dot{I}}_{sbref}$$

To ensure the exponential convergence of $I_{sb}$ to its reference value, the forced dynamic behavior of $\varepsilon_{3}$ is as follows.
(51)$${\dot{\varepsilon }}_{3}=-d_{3}\varepsilon_{3}$$

By combining Equations (50) and (51), the control bidirectional DC-DC converter is obtained.
(52)$$M_{23}=\frac{L_{2}}{U_{dc}}\left[d_{3}\varepsilon_{3}+\frac{U_{sb}}{L_{2}}-{\dot{I}}_{sbref}\right]$$

### 3.4. Energy Management

The goal of energy management is to effectively manage the energy transfer flow between the PV generator, batteries, and load. In fact, when the electric vehicle is located in a home garage or a covered area, the solar radiation remains insufficient to supply the needed power for starting the vehicle. The demand power is to be provided by a storage battery.

Let us design with switches K1, K2, and K3—the used switches that supervise the energy transfer flow. Switch K1 supervises the transfer of energy for the delivered PV generator energy to the load only. Switch K2 is used to control the transfer of energy between the PV generator and the battery only. Finally, switch K3 is used to supervise the transfer of energy between the battery and the load only.

The decision parameters of the energy management are the power delivered by the PV generator, the battery state of charge (SOC), and the demanded power load.

The main objectives of the power management algorithm are to extract maximum power from the PV generator, avoid overcharge and deep discharge in the battery, and assume the load energy demands.

Depending on demand, the PV generator’s produced energy, and the battery SOC, the system operates in one of the following cases. Taking into account the complexity time Tc, the conceived management algorithm is shown in Algorithm 1.
**Algorithm 1** The power management algorithmCompute starting time Tb
Repeat{{ if (vehicle will start)The vehicle is totally supplied with the batteryelse (vehicle is in motion)Extract the maximum power from the PV generatorif (weather is sunny)if (produced PV power exceeds the required load power)if (the battery is fully charged){-Disconnect the battery-The load is supplied with the PV generator}if (the battery ready to charge){-Charge the battery-Supply the load}if (the battery is ready to discharge){-Supply the load with the produced PV power-Offset the lack of load energy by the battery stored energy}if (the battery is fully discharged){-Supply the load-Disconnect the battery}elseif (the battery is fully charged){{-Supply the load with the produced PV power-Offset the lack of load energy by the battery stored energy}else-Disconnect the battery}}Compute the end time Tend}Until (Tend−Tb≥Tc)

## 4. Simulation Results and Discussion

The performance of the robust optimization and energy management strategy based on nonlinear controllers for electric vehicles is highlighted by means of numerical simulations.

### 4.1. System Characteristics

The specifications of the used electric vehicle in simulation are given in Table 3.

The mechanical and electrical characteristics of the used electric three-phase PMSM motor are summarized in Table 4.

The used PV generator consists of triple-junction solar panels. It provides 100 kw at standard test conditions of 1000 w/m^2^ and T = 25 °C.

The parameters of the triple-junction InGap/InGaAs/Ge solar cell are shown in Table 5.

The battery storage bank is obtained by the association of 84 Panasonic Lithium-ion CGR18650E battery cells in series and 40 Panasonic Lithium-ion CGR18650E battery cells in parallel. The following characteristics of the used battery cell are regrouped in Table 6.

### 4.2. Behavior Energy Management and Nonlinear Controllers’ Efficiency

In this section, the aim is to verify the performance of the conceived controllers and to testify to the validity of the energy management strategy. Different operating and environmental conditions are considered.

#### 4.2.1. Case of Quick Response

In order to validate the performance of the conceived algorithms and the energy management strategy at quick response, specific trajectories for both operating and meteorological conditions are considered. In fact, as shown with Figure 11e, the target vehicle speed is fixed to zero value from *t* = 1 s to *t* = 2 s. Since *t* = 2 s, the speed quickly increase to 40 km/h. The used radiation and temperature trajectories, in this case, are represented in Figure 11a,b, respectively. As it is indicated with these figures, radiation and temperature are both fixed to 900 w/m^2^ and 60 °C, respectively. Since *t* = 2 s, they simultaneously increase to 950 w/m^2^ and 70 °C, respectively. From *t* = 4 s to *t* = 6 s, radiation increases to 1000 w/m^2^ and the temperature decreases to 50 °C. Finally, *t* = 6 s, a 11,000 w/m^2^ is associated to radiation and the temperature is fixed to 64 °C. The evolution of the available PV generator and its optimal one is given in Figure 11c. As it is indicated in this figure, the available PV generator precisely and rapidly tracks its optimal value despite simultaneous abrupt alteration in radiation, and temperature and load variation (Figure 11f).

The validity of the energy management algorithm is also noticed from the obtained results. In fact, the evolution of the DC-link voltage is given in Figure 11d. Despite the changes in radiation, in temperature, and in load torque, DC-link voltage is maintained constant, except some fluctuation appeared at the time disturbances variation. The battery state of charge and the working modes are, respectively, depicted in Figure 11g,h. Suitable charging and discharging modes are shown over the evolution of the battery state of charge. This is proved with the system operating modes proving the validity of the used energy management strategy.

#### 4.2.2. Case of Variable Vehicle Speed Response

Here, the aim is to verify the tracking behavior of the conceived controllers and to testify to the validity of the energy management strategy under internal and external disturbances for a variable speed operation including both normal and field-weakening operating modes. The adapted temperature and radiation trajectories are shown in Figure 12a. A suitable target speed trajectory for electric vehicle applications, including different operating conditions, is used as reported in Figure 12b. Figure 12c shows that the PV generator rapidly tracks its maximum values despite abrupt meteorological conditions and abrupt load variations (Figure 12a,d). This will significantly reduce the recharge time. As it is indicated in Figure 12e, the DC–link voltage is maintained fixed at its target value, with some fluctuations caused by the load and meteorological variations. Taking into account the vehicle power (Figure 12f) and the battery state of charge (Figure 12g), the states of the switches K_1_, K_2_, and K_3_ (Figure 12i, j and k) and the working modes are obtained (Figure 12h). The obtained results show that the energy management approach is working well.

By using the conceived energy management algorithm, the protection of the battery is assumed. In fact, as it is shown in Figure 12g, the SOC is always maintained between the maximum and minimum value of SOC.

#### 4.2.3. Case of Extra Urban Drive Cycle (EUDC) Response

In this simulation test, the EUDC is used for the target vehicle speed (Figure 13g) under simultaneous abrupt radiation and temperature variations. The type of the road in which the electric vehicle moves and the impact of the wind are also considered. Therefore, a simultaneous quick change in radiation and temperature is used. The adopted radiation and temperature trajectories are, respectively, plotted in Figure 13a,b. A vehicle road, including slope inclination, is used as shown in Figure 13c. In fact, the vehicle road is inclined with two slope angles. The first is from *t* = 2 s to *t* = 4 s and the second is applied at *t* = 6 s to *t* = 8 s. A random trajectory for the wind speed is chosen in simulation test, as illustrated in Figure 13d. Available PV generator power sufficiently tracks its optimal power (Figure 13e) despite atmospheric conditions, the type of road, and load torque variation (Figure 13h). The validity of the power management tacking into account the battery safety is effectively highlighted, as it is shown with the battery state of charge (Figure 13i), the DC-link voltage (Figure 13f), and the operating modes (Figure 13j).

## 5. Conclusions

In this paper, an improved energy management approach for electric vehicle is designed. The electric vehicle structure is composed of a triple junction PV generator, a lithium-ion battery storage, DC-DC converters, and an electric vehicle. All necessary equations are given for each of the electric vehicle elements. A nonlinear robust MPPT algorithm is designed and applied in the PV generator optimization. For the DC-DC converters, Lyapunov function-based nonlinear controllers are conceived. Moreover, a based rule approach is designed for the energy management. To show the performance of the conceived energy management and the control algorithms, the global model of the electric vehicle and the designed algorithms are implemented and validated in the Matlab/Simulink platform.

The obtained results show that designed models are operated well. The control algorithms are tested under different working conditions. The control objectives are actually met by using the developed algorithm, and the electric vehicle recharge time is improved. The weight of the electric vehicle is also improved by using the multi-junction solar cell technology. This fact leads to minimizing the energy consumption.

In the energy management approach, at the case of energy excess and when the battery is fully charged, the battery is disconnected. PV generator fully supplies the electric vehicles. To improve the electric vehicle performance, a new management strategy that takes into account the protection of the traction part of electric vehicle is to be developed. This issue is one of the work’s future prospects.

## Figures and Tables

**Figure 1 sensors-22-06123-f001:**
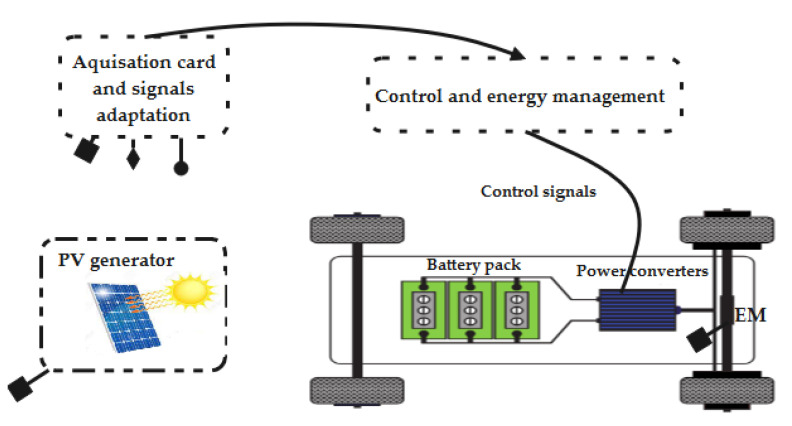
A description of the studied system.

**Figure 2 sensors-22-06123-f002:**
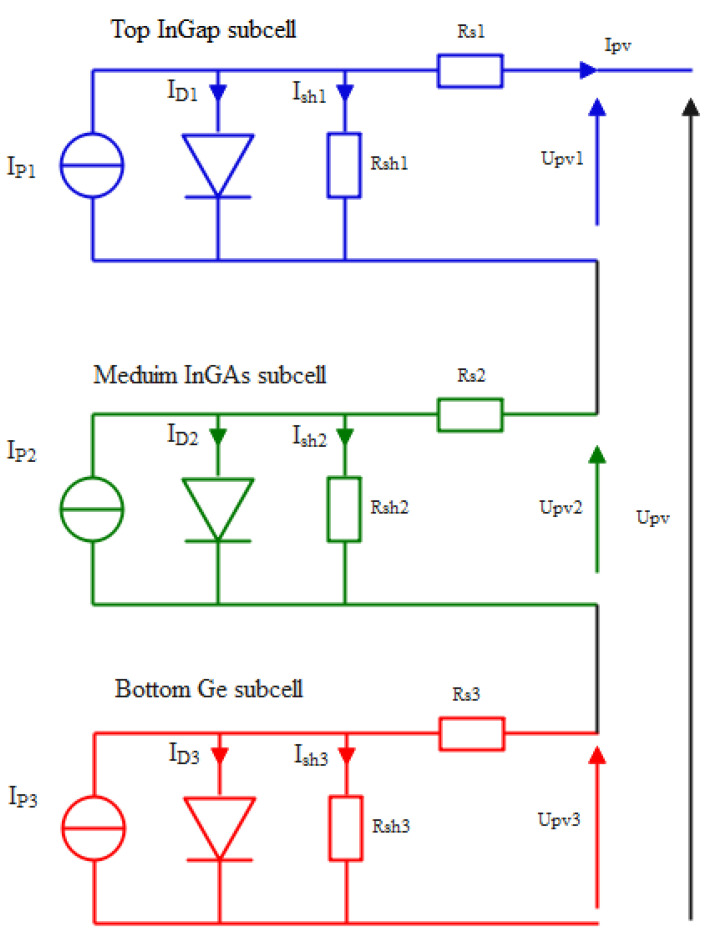
Electrical equivalent circuit of a multi-junctio InGap/InGaA/Ge solar cell.

**Figure 3 sensors-22-06123-f003:**
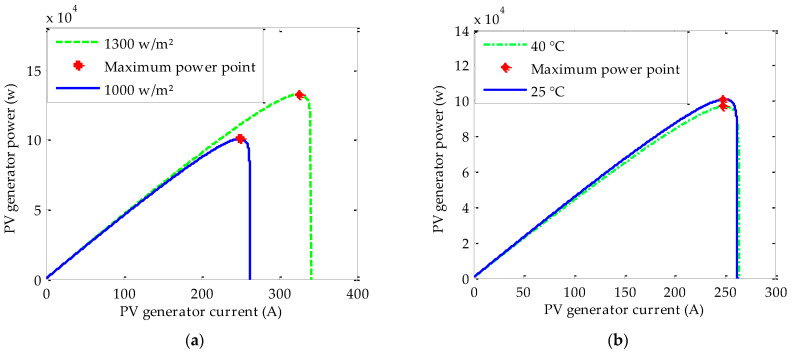
PV generator $P_{pv}=f(V_{pv})$ characteristic: (**a**) under radiation variation and (**b**) under temperature variation.

**Figure 4 sensors-22-06123-f004:**
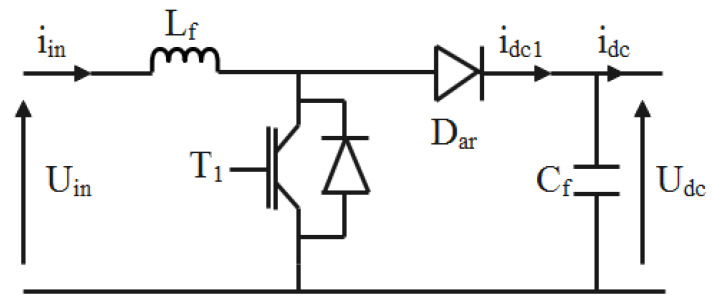
Unidirectional boost chopper.

**Figure 5 sensors-22-06123-f005:**
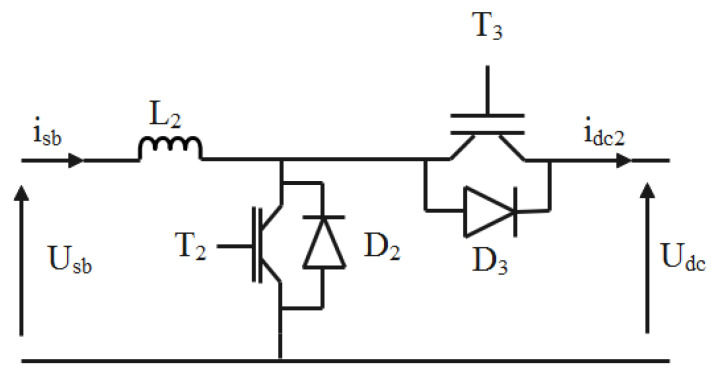
Bidirectional buck-boost chopper.

**Figure 6 sensors-22-06123-f006:**
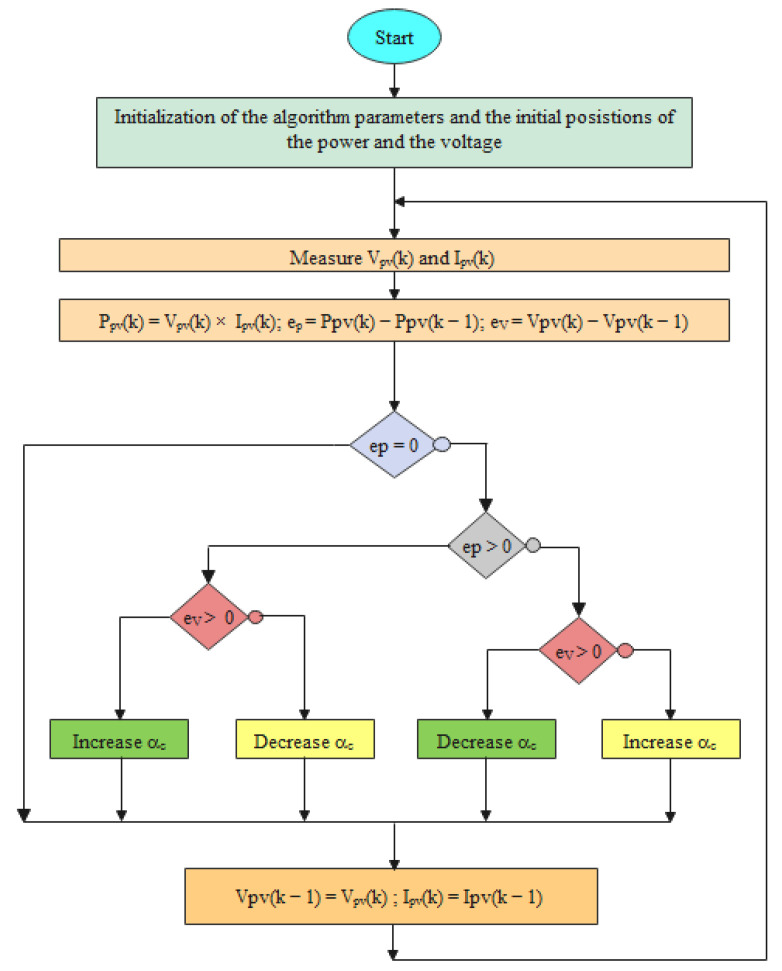
Flowchart of the P & O MPPT algorithm.

**Figure 7 sensors-22-06123-f007:**
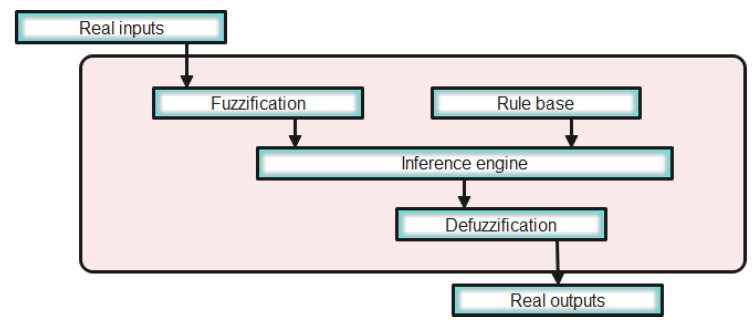
Schematic of the mamdani Fuzzy logic system architecture.

**Figure 8 sensors-22-06123-f008:**
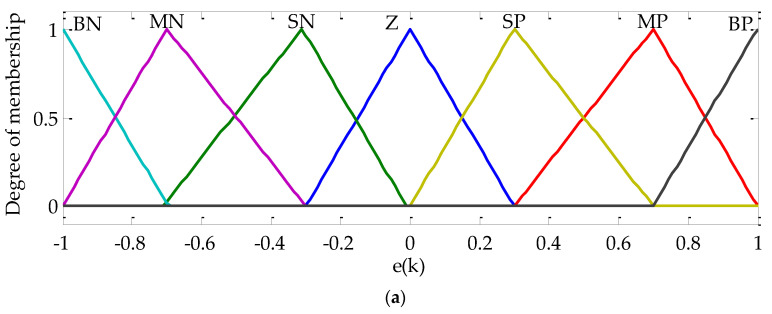
Fuzzy logic MPPT algorithm membership functions: (**a**) Error membership function, (**b**) Change error membership function, and (**c**) change duty cycle membership function.

**Figure 9 sensors-22-06123-f009:**
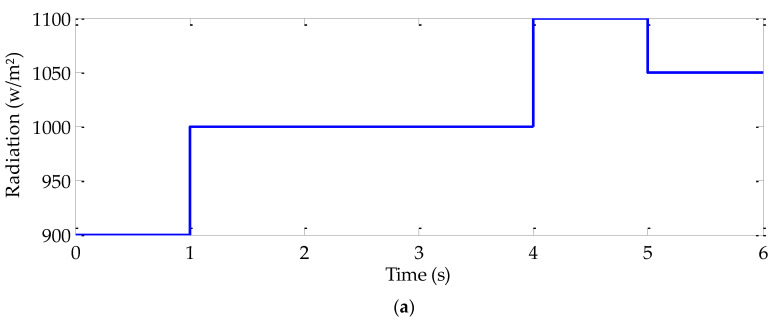
Simulation results given by the three MPPT algorithms under various operating conditions: (**a**) Radiation trajectory, (**b**) Temperature trajectory, (**c**) Load resistor trajectory, (**d**) Duty cycles, and (**e**) PV generator powers.

**Figure 10 sensors-22-06123-f010:**
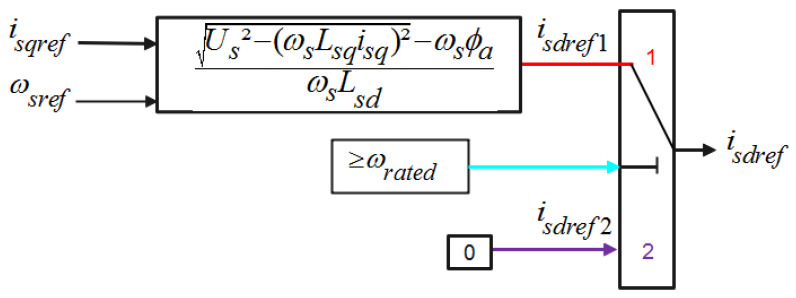
High-speed control algorithm.

**Figure 11 sensors-22-06123-f011:**
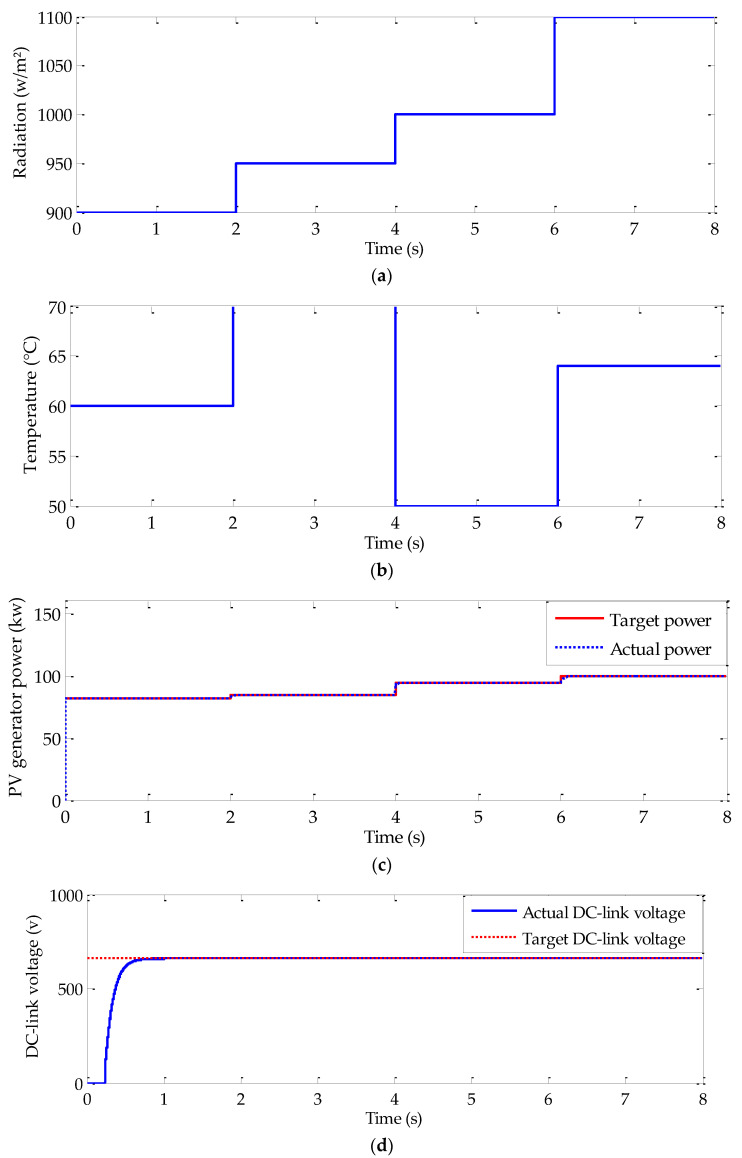
Simulation results under quick response: (**a**) radiation trajectory, (**b**) temperature trajectory, (**c**) PV generator power, (**d**) DC-link voltage, (**e**) vehicle speed, (**f**) load torque, (**g**) battery state of charge, and (**h**) operating mode.

**Figure 12 sensors-22-06123-f012:**
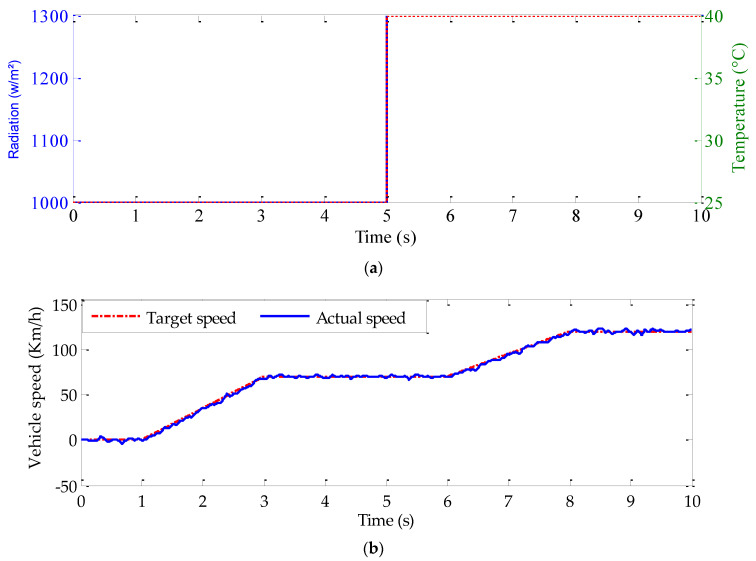
Simulation results under variable target vehicle speed: (**a**) temperature and radiation trajectories, (**b**) vehicle speeds, (**c**) PV generator power, (**d**) load torque, (**e**) DC–link voltage, (**f**) vehicle consumed power, (**g**) battery state of charge, (**h**) battery state of charge operating mode, (**i**) switch K1, (**j**) switch K2, and (**k**) switch K3.

**Figure 13 sensors-22-06123-f013:**
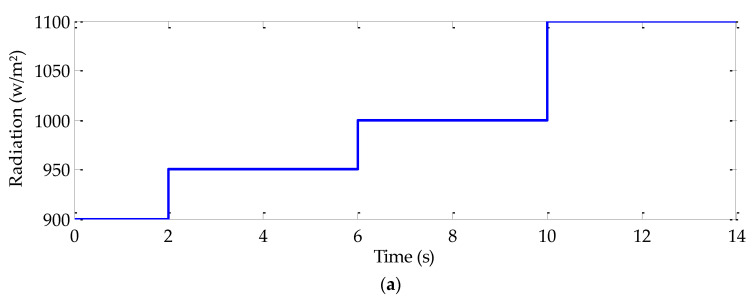
Simulation results under extra urban drive cycle response: (**a**) radiation trajectory, (**b**) temperature trajectory, (**c**) street inclination angle, (**d**) wind speed, (**e**) PV generator power, (**f**) DC-link voltage, (**g**) vehicle speed, (**h**) load torque, (**i**) battery state of charge, and (**j**) operating modes.

**Table 1 sensors-22-06123-t001:** Rule-based table of the fuzzy logic MPPT algorithm.

Δu(k)		Δe(k)						
		BN	MN	SN	Z	SP	MP	BP
e(k)	BN	Z	SN	MN	BN	BN	BN	BN
	MN	SP	Z	SN	MN	BN	BN	BN
	SN	MP	SP	Z	SN	MN	BN	BN
	Z	BP	MP	SP	Z	SN	MN	BN
	SP	BN	BP	MP	SP	Z	SN	SN
	MP	BP	BP	BP	MP	SP	Z	SN
	BP	BP	BN	BP	BP	MP	SP	Z

**Table 2 sensors-22-06123-t002:** Comparative study between the three MPPT algorithms.

				MPPT Algorithms			
Operating Modes	G (w/m^2^)	T (°C)	R (Ohm)	Parameters	SMC	FL	P and O
Simultaneous radiation and temperature variation	900	27	5	Response time (ms)	8.75	100	313.8
				Tracking error (%)	−5.3 × 10^−6^	5.2 × 10^−5^	2.6 × 10^−4^
				Objective function value	0.006	0.01	−0.25
	1000	77	5	Stabilization time (ms)	52	105	224
				Tracking error (%)	9.12 × 10^−8^	8.6 × 10^−5^	1.73 × 10^−5^
				Voltage loss (%)	1.5 × 10^−3^	6 × 10^−3^	5.5 × 10^−3^
				Objective function value	0.008	0.048	0.09
Abrupt load variation	1000	77	10	Stabilization time (ms)	≅0	137	416
				Tracking error	1.75 × 10^−5^	1.6 × 10^−4^	3.7 × 10^−3^
				Voltage loss (%)	2 × 10^−4^	16 × 10^−4^	117 × 10^−4^
				Objective function value	−2.3 × 10^−4^	−0.091	0.3
	1000	77	15	Stabilization time (ms)	≅0	69	125
				Tracking error (%)	2.1 × 10^−4^	2.7 × 10^−4^	8 × 10^−4^
				Voltage loss (%)	5 × 10^−4^	10 × 10^−4^	8.9 × 10^−3^
				Objective function value	0.01	0.011	0.02
Simultaneous radiation, temperature, and load variation	1100	87	8	Response time (ms)	44	58	144
				Tracking error (%)	−4.27 × 10^−7^	1.9 × 10^−6^	2.9 × 10^−4^
				Voltage loss (%)	1.3 × 10^−3^	10 × 10^−3^	14.5 × 10^−3^
				Objective function value	3.6 × 10^−4^	−0.07	8.4 × 10^−4^
	1050	52	12	Response time (ms)	≅0	35	0.2
				Tracking error (%)	2.6 × 10^−5^	1.1 × 10^−4^	22.7 × 10^−4^
				Voltage loss (%)	5.4 × 10^−4^	1.6 × 10^−3^	3.3 × 10^−3^
				Objective function value	6.8 × 10^−3^	0.03	−0.1992
Number of controllers tuning parameters					01	03	01

**Table 3 sensors-22-06123-t003:** Electric vehicle parameters.

Parameter	Value
Vehicle mass (kg)	1450
Vehicle frontal area (m^2^)	2.711
Tire radius (m)	0.43
Aerodynamic drag coefficient	0.29
Air density (kg/m^3^)	1.204
rolling resistance coefficient	0.013

**Table 4 sensors-22-06123-t004:** 100 kw PMSM parameters.

Parameter	Value
Direct inductance (mH)	0.17
Reverse inductance (mH)	0.29
Flux linkage (wb)	0.071
Stator resistance (Ω)	0.0083
Number of poles	8
Viscous friction (Nm/rad/s)	0.005
Moment of inertia (kg m^2^)	0.089

**Table 5 sensors-22-06123-t005:** Triple-junction InGap/InGaAs/Ge solar cell parameters.

	Top Sub-Cell InGaP	Top Sub-Cell InGaAs	Top Sub-Cell Ge
Band-gap energy (ev)	1.976	1.519	0.744
Short circuit current (A)	6.7522	7.7126	10.094
Diode ideality factor	1.97	1.75	1.96
$K_{i}$ (A/cm^2^k^4^)	1.86 × 10^−9^	1.288 × 10^−8^	10.5 × 10^−6^
$\delta_{i}$	2	2	2
$\alpha_{i}$	7.5 × 10^−4^	5.405 × 10^−4^	4.774 × 10^−4^
$\beta_{i}$	500	204	235

**Table 6 sensors-22-06123-t006:** Panasonic Lithium-ion CGR18650E battery cell parameters.

Parameter	Value
$Q_{n}$ (Ah)	2.55
$K_{sb}$ (v)	0.0152
$A_{b}$ (v)	0.071
$B_{b}$ (Ah^−1^)	2.893
$Ri_{n}$ (Ω)	0.1138

## Data Availability

Not applicable.

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
