# Peer review of "Robust Optimization and Power Management of a Triple Junction Photovoltaic Electric Vehicle with Battery Storage"

_sensors, 2022, doi:10.3390/s22166123_

Round 1
Reviewer 1 Report
The simulation part should be enhanced by adding a simulation scenario showing how the vehicle is tested based on a specific path within certain operating and environment conditions.
In the power management algorithm, it is better to link the different rules and their firing with the load type and the operating conditions. For example, based on the vehicle type and the task it is going to do (transportation) and the weather in which it moves, a specific rule may be used.
A performance analysis should be applied for the whole integrated algorithm to investigate its time and space complexity. Especially, a quick response may be needed at specific conditions.
The quality of Figures 1 and 6 are to be improved.
A quantitative comparison with related work in literature should be given.
Author Response
Dear Honorable Reviewer,
Greetings!
The authors express their sincere acknowledgement and thanks to you for devoting your valuable time to reviewing the manuscript and providing necessary comments and suggestions. These suggestions and comments have undoubtedly contributed to an improvement in the overall quality of the paper. The authors have taken the comments very positively and have tried to answer each comment in the best possible way. The comments and suggestions of the honorable editor and reviewers have been incorporated into the revised manuscript and are highlighted.
The pointwise replies to all the comments/queries are submitted for kind perusal and are attached.

Reviewer 2 Report
The paper presents an approach for improving EV performance. Unfortunately, I cannot recommend publishing this work at the moment due to the following reasons.
1. Dumping of references have to be avoided. For one line of literature 3-4 references have been used. For example, in [1-4], [5-8], [19-23] and the worst [24-32] which are 8 references for 3 lines of literature. Please explain them all or limit to 1 or 2 relevant references to make your point. Artificially, adding literatures for its sake is unacceptable.
2. Even though in the first page Sensors is used, from page 2 in the header Electronics journal is used.
3. The algorithm is incorrect. While the logic is okay the use of else and elseif have to be corrected. It is hard to understand the flow of decision making.
4. In the algorithm If PV power > load power is explained also should be explained load power > PV power. I think it is similar to the "elseif (the battery is ready to discharge)" but still that case has to be included in the right manner within the algorithm.
5. Robust optimization has not been explained.
6. The optimization model including objective functions, constraints etc has not been explained. Only a general text has been included
" The decision parameters of the energy management are the power delivered by the 312 PV generator, the battery state of charge (SOC), and the demanded power load. 313 The main objectives of the power management algorithm are to extract maximum 314 power from the PV generator, avoid overcharge and deep discharge in the battery, and 315 assume the load energy demands"
does this mean that a multi-objective function has been considered ? what are the equality/inequality constraints in this case ?
Author Response

(The authors gave the same response as above.)

Round 2
Reviewer 1 Report
It is still not clear how complex is the implemented power management algorithm. In some related scenarios and applications, the time needed by the algorithm to finalize computation and generate actions (time complexity), should be taken into consideration. It can be investigated as lag or latency in the system.
Author Response
Dear Respected Reviewer
Thanks you once again your constructive comments and suggestions. These suggestions and comments have undoubtedly contributed to improvement in the overall quality of the paper. The authors have taken the comments very positively and have tried to answer each comment in the best possible way.
The pointwise replies of all the comments/queries are attached here for kind perusal:

Reviewer 2 Report
The authors have significantly improved the paper. Most of my concerns have been addressed and the paper has improved in quality suitable for publication.
Author Response

(The authors gave the same response as above.)
